# Evolutionary Dynamics of Whole-Genome Influenza A/H3N2 Viruses Isolated in Myanmar from 2015 to 2019

**DOI:** 10.3390/v14112414

**Published:** 2022-10-31

**Authors:** Wint Wint Phyu, Reiko Saito, Yadanar Kyaw, Nay Lin, Su Mon Kyaw Win, Nay Chi Win, Lasham Di Ja, Khin Thu Zar Htwe, Thin Zar Aung, Htay Htay Tin, Eh Htoo Pe, Irina Chon, Keita Wagatsuma, Hisami Watanabe

**Affiliations:** 1Division of International Health (Public Health), Graduate School of Medical and Dental Sciences, Niigata University, Niigata 951-8510, Japan; 2Infectious Diseases Research Center of Niigata University in Myanmar (IDRC), Graduate School of Medical and Dental Sciences, Niigata University, Niigata 951-8510, Japan; 3Respiratory Medicine Department, ThingangyunSanpya General Hospital, Yangon 110-71, Myanmar; 4Microbiology Section, (200) Bedded Pyinmana General Hospital, Naypyitaw 150-31, Myanmar; 5Department of Microbiology, University of Medicine, Mandalay 050-21, Myanmar; 6Microbiology Section, Mandalay General Hospital, Mandalay 050-31, Myanmar; 7National Health Laboratory, Department of Medical Services, Dagon Township, Yangon 111-91, Myanmar; 8Japan Society for the Promotion of Science, Tokyo 102-0083, Japan

**Keywords:** seasonal influenza, A/H3N2, next-generation sequencing, evolution, whole-genome sequencing, mutation, variant viruses, genetic reassortment

## Abstract

This study aimed to analyze the genetic and evolutionary characteristics of the influenza A/H3N2 viruses circulating in Myanmar from 2015 to 2019. Whole genomes from 79 virus isolates were amplified using real-time polymerase chain reaction and successfully sequenced using the Illumina iSeq100 platforms. Eight individual phylogenetic trees were retrieved for each segment along with those of the World Health Organization (WHO)-recommended Southern Hemisphere vaccine strains for the respective years. Based on the WHO clades classification, the A/H3N2 strains in Myanmar from 2015 to 2019 collectively belonged to clade 3c.2. These strains were further defined based on hemagglutinin substitutions as follows: clade 3C.2a (*n* = 39), 3C.2a1 (*n* = 2), and 3C.2a1b (*n* = 38). Genetic analysis revealed that the Myanmar strains differed from the Southern Hemisphere vaccine strains each year, indicating that the vaccine strains did not match the circulating strains. The highest rates of nucleotide substitution were estimated for hemagglutinin (3.37 × 10^−3^ substitutions/site/year) and neuraminidase (2.89 × 10^−3^ substitutions/site/year). The lowest rate was for non-structural protein segments (4.19 × 10^−5^ substitutions/site/year). The substantial genetic diversity that was revealed improved phylogenetic classification. This information will be particularly relevant for improving vaccine strain selection.

## 1. Introduction

Influenza viruses cause acute respiratory illnesses that can affect people of all ages [1]. Every year, 5–20% of the human population is infected with influenza viruses. Globally, these infections are a major cause of human morbidity, with an estimated 3–5 million cases of severe illness each year and mortality of approximately 290,000 to 650,000 deaths each year [2]. Data were collected before the COVID-19 pandemic. In industrialized countries, most deaths associated with influenza occur in people aged ≥65 years [3]. Nearly 99% of deaths in children under 5 years of age owing to influenza-related lower respiratory infections occur in developing countries [4].

Influenza viruses are negative-sense, single-stranded, enveloped RNA viruses that belong to the Orthomyxoviridae family. Their genome comprises eight gene segments: RNA polymerase subunits polymerase basic 2 (PB2), polymerase basic 1 (PB1), and polymerase acidic (PA), which encode viral RNA-dependent RNA polymerase and other proteins that have been proposed to induce cell death (e.g., PB1–F2) and modulate viral pathogenicity (e.g., PA-X^9^), hemagglutinin (HA), nucleoprotein (NP), neuraminidase (NA), matrix (M), and non-structural protein (NS) [5]. There are four types of Influenza viruses: A, B, C, and D [2]. Influenza A and B viruses circulate and cause seasonal epidemics [2]. Influenza A viruses are further classified into 18 HA subtypes (H1–H8) and 11 NA subtypes (N1–N11) based on the reactivity of their surface HA and NA glycoproteins, with a total of 144 subtypes possible. In contrast, the influenza B virus has no subtypes [5,6,7,8]. The influenza A viruses currently circulating in humans are subtypes A/H1N1 and A/H3N2.

There was a large outbreak of influenza A/H3N2 during the 2010–2015 season in Myanmar [9]. Multiple influenza types/subtypes usually circulate during each season in Myanmar; for example, A/H1N1pdm was predominant in 2010 and 2015, A/H3N2 was the predominant circulating virus in 2011, and influenza B viruses were the predominant viruses in 2012 and 2014 [9]. The first case of an oseltamivir-resistant strain was detected in influenza A/H1N1pdm viruses circulating in Myanmar in 2017 [10].

Influenza A viruses evolve rapidly through both mutation and reassortment. Within each virus subtype, the gradual accumulation of nucleotides and amino acid substitutions in HA and NA surface glycoproteins periodically results in the emergence of new antigenic variants, a phenomenon called antigenic drift [11]. New antigenic variants of A/H3N2 viruses appear every 3–5 years, whereas new antigenic variants of A/H1N1 and influenza B viruses appear less frequently (3–5 years for A/H3N2 viruses compared to 3-8 years for A/H1N1 and influenza B viruses) [11,12,13]. Influenza A/H3N2 viruses are thought to have evolved faster than other subtypes [14,15]. Recent studies have revealed extensive clade diversity and increased morbidity and mortality rates [16,17]. During influenza epidemics from 2016 to 2017 and 2017 to 2018, the majority of the predominant influenza viruses in the Southern Hemisphere were A/H3N2 viruses in clade 3C.2 [18,19]. The segmented nature of the genome allows genetic reassortment when a cell is coinfected by two or more strains resulting in rapid antigenic drift. Inter-subtype reassortment gives rise to viruses with pandemic potential [20].

Next-generation sequencing (NGS) technologies are increasingly used for molecular surveillance of influenza viruses in many countries [21,22,23]. As the role of intra-subtype reassortment becomes increasingly clear, identifying reassortments with whole-genome-based surveillance is the gold standard tool, especially in the case of recombinogenic viruses, such as influenza.

Influenza vaccine effectiveness can vary depending on the antigenic match between circulating virus strains and vaccine components, the time since vaccination because of waning immunity, and host factors such as age, immune function, earlier exposure to influenza viruses, and vaccination history [24,25,26]. Vaccine effectiveness might also be reduced by molecular changes arising from egg adaption for vaccines that use viruses grown in embryonated eggs, such as the influenza A/H3N2 vaccine component [27].

We previously reported multiple influenza types/subtypes circulating during the 2010–2015 season in Myanmar and performed a genetic analysis of HA to assess the transmission pathway of influenza viruses between Myanmar and other countries [9]. However, data on the whole genome of the influenza virus are generally sparse in Myanmar. Therefore, we aimed to study the whole-genome sequences of influenza A/H3N2 viruses circulating in Myanmar spanning the seasons from 2015 to 2019 in comparison with the World Health Organization (WHO)-recommended Southern Hemisphere vaccine strains of the respective years to detect the emergence of new variants of the influenza virus. A better understanding of whole-genome sequences can assist in the exploration of inter- and intra-seasonal evolutionary dynamics and the identification of mutations located anywhere in the genome. This knowledge will improve the classification and detection of reassortments.

## 2. Materials and Methods

### 2.1. Sample Collection

Sudden onset of symptoms, including fever and respiratory symptoms such as cough, and rhinorrhea, define influenza-like illness (ILI). Nasopharyngeal swabs were collected from outpatients with ILI who visited the Thinganguyn Sanpya General Hospital in Yangon and the 200-bed General Hospital in Pyinmana, Myanmar, from January 2015 to December 2019. Patient information, including sex, age, and clinical symptoms, was also recorded. Samples were collected after obtaining written informed consent from the patients or guardians of all study participants. Two samples were collected for each participant. The first sample was screened for influenza using the Quick-Navi Flu+RSV rapid diagnostic test (Denka Seiken, Tokyo, Japan) at the sample collection sites. The second sample was placed in a viral transport medium and frozen at −20 °C at the study sites. Within a few months of sample collection, the samples were transported by an international courier to the Division of International Health (Public Health), Graduate School of Medical and Dental Sciences, Niigata University, Niigata, Japan, for laboratory examination. This study was approved by the Niigata University Ethical Committee (2015-2533) and the Ethical Review Committee of the Department of Medical Research, Ministry of Health and Sports, Myanmar (No. 016516).

### 2.2. Cells

Madin–Darby Canine Kidney (MDCK) cells were maintained in minimum essential medium (MEM; Sigma-Aldrich, St. Louis, MO, USA), supplemented with 10% fetal bovine serum (FBS; Nichirei Biosciences Inc., Tokyo, Japan), 100 U/mL penicillin, and 100 μg/mL streptomycin (Gibco, Cambridge, MD, USA). MDCK cells stably expressing human 2,6-sialyltransferase (MDCK-SIAT1) were maintained in Dulbecco’s modified Eagle’s medium (DMEM; Sigma) supplemented with 10% FBS (Nichirei Biosciences Inc.), 2 mM L-glutamine (Wako, Osaka, Japan), and 1.0 mg/mL G418 (Promega, Madison, WI, USA) [28]. All cell lines were cultured at 37 °C in a 5% CO_2_ humidified incubator.

### 2.3. Viral Isolation

For viral isolation using MDCK cells, the clinical specimens from nasal swabs were diluted to 1.5 times in MEM containing 35 µg/mL trypsin (Sigma), 1 × MEM vitamin solution (Gibco), D-glucose (Sigma), 100 U/mL penicillin and 100 μg/mL streptomycin (Gibco). For viral isolation using MDCK-SIAT1 cells, clinical specimens from nasal swabs were diluted 1.5 times in DMEM containing 2 mM L-glutamine (Wako), 0.1% bovine serum albumin (Sigma), 1 µg/mL TPCK trypsin (Sigma), 100 U/mL penicillin and 100 μg/mL streptomycin (Gibco) [28]. Starting in 2018, we began to use MDCK-SIAT1 cells instead of MDCK cells for virus isolation; MDCK-SIAT1 cells showed improved isolation rates for more recent influenza viruses, including H3N2, compared to conventional MDCK cells [29]. MDCK and MDCK-SIAT1 cells were seeded into 48-well plates and incubated at 37 °C and 5% CO_2_. The next day, the cells were inoculated with the diluted clinical specimens and incubated at 34 °C and 5% CO_2_. After inoculation, cells were monitored for cytopathic effects for 3–10 days [30]. The samples underwent a second passage using the same cells if no cytopathic effects were observed during the first inoculation. After 3–10 days of incubation, the culture supernatants and cells were collected and stored at −80 °C. A mixture of the culture supernatants and cells was used for RNA extraction. The presence of the influenza virus in the supernatant was confirmed by quantitative real-time polymerase chain reaction (qPCR) [28].

### 2.4. Sample Selection

For genetic characterization of the whole-genome sequences of influenza, 79 A/H3N2 amplified polymerase chain reaction (PCR) products were selected from archived viruses, with a cycle threshold (C_t_) value ≤ 32, as detected by qPCR, and sampling was performed on different months and dates of sample collection.

### 2.5. RNA Extraction and PCR Amplification for NGS

Viral nucleic acids were extracted using a High Pure Viral RNA Kit (Roche Diagnostics, Mannheim, Germany) according to the manufacturer’s instructions. Sequencing amplicons were generated using one-step reverse transcription PCR (PrimeScript II High Fidelity; TaKaRa Bio, Shiga, Japan). A 20-µL reaction mixture containing 1 µL eluted RNA and final concentrations of 2 × Primescript II one-step High Fidelity buffer, 0.4 µL PrimeScript II High Fidelity Enzyme Mix, and 1.6 µL PrimeSTAR GXL was prepared. Reverse transcription–qPCR amplification was performed using three primers: 5 µM common-uni 12R (5′-GCCGGAGCTCTGCAGATATCAGCRAAAGCAGG-3′), 5 µM common-uni 12G (5′-GCCGGAGCTCTGCAGATATCAGCGAAAGCAGG-3′), and 0.4 µM common-uni13 (5′-CAGGAAACAGCTATGACAGTAGAAACAAGG-3′), allowing reverse transcription and amplification of each segment. This established protocol [31] uses modified primers containing a mixture of common-uni 12R and common-uni 12G primers, as well as 0.4 µM common-uni13 primers. The thermal cycling conditions were: reverse transcription at 45 °C for 10 min; initial denaturation/enzyme activation of 94 °C for 2 min; 35 cycles of 98 °C for 10 s, 60 °C for 15 s; and final extension of 68 °C for 30 s, followed by storage at 4 °C.

### 2.6. Library Preparation and Sequence Runs for NGS

Amplified products were purified using a Wizard SV Gel and PCR Purification Kit (Promega, Madison, WI, USA), according to the manufacturer’s instructions. Purified products were examined with an Agilent 4200 TapeStation (Agilent Technologies, San Diego, CA, USA) using a D 1000 ScreenTape system. The concentration of each purified product was quantified with a Qubit Flex Fluorometer (Thermo Fisher Scientific, Waltham, MA, USA) using a Qubit-iT 1×dsDNA HS assay (Thermo Fisher Scientific). Each sample was diluted to a DNA concentration of 50 ng/µL, and the purified RT-PCR products were used to prepare sequencing libraries using a Nextera Flex DNA sample preparation kit (Illumina, San Diego, CA, USA) before sequencing. All libraries were sequenced on an iSeq 100 platform (Illumina, San Diego, CA, USA). The amplicons were sheared to lengths between 200 and 400 bp, and Illumina sequencing adapters were ligated to the sheared DNA using Nextera Flex (Illumina). The libraries were multiplexed, clustered, and sequenced using a 300-cycle (2 × 150 bp paired-end) ISeq v2 reagent kit (Illumina on the iSeq 100 platform according to the manufacturer’s protocols. NGS was used to obtain the nucleotide sequences of all eight viral segments.

### 2.7. NGS Data Analysis

Raw sequences (paired-end) approximately 250 bp in length were trimmed. Contigs for individual gene segments from influenza viruses were generated using the de novo assembly module of CLC Genomics Workbench bio software v.20.0.2 (CLC bio, Cambridge, MA, USA).

### 2.8. Subclade Classification by Amino Acid Substitutions in HA

All sequence reads were mapped to the selected reference strains after retrieval using the CLC biosoftware v.20.0.2. As reference strains, the WHO-recommended Southern Hemisphere vaccine strains from 2014 to 2020 were downloaded from the influenza research database (https://www.fludb.org/brc/home.spg?decorator=influenza; accessed on 12 June 2021). The subclades of the 79 A/H3N2 in this study were determined by the key amino acid substitutions in HA that define the subclades of A/H3N2 as proposed by the WHO (https://www.crick.ac.uk/research/platforms-and-facilities/worldwide-influenza-centre/annual-and-interim-reports; accessed on 28 April 2022). Key amino acid substitutions based on A/Texas/50/2012, a Southern Hemisphere vaccine strain in 2014, were used to classify the HA 3C.2a group comprising 3C.2a1, 3C.2a1b, and 3C.2a2. Subclassification of the vaccine strains for each year was performed based on the HA substitutions. Genetic analysis of the amino acid substitutions was performed using the MEGA software v.7.0.26 (https://www.megasoftware.net/; accessed on 12 June 2021).

To better understand the clade classification and group clustering of the viruses, a phylogenetic tree was constructed using 12 strains from India and Thailand after comparing them with the Southern Hemisphere vaccine strains of known clades recommended by the WHO. Moreover, to understand the genetic relationship between the Myanmar strains and globally circulating strains, we also performed a BLAST search for the HA gene of 79 A/H3N2 sequences from this study and constructed a phylogenetic tree.

### 2.9. Evaluation of Genetic Match with Vaccine Strains

Whole genomes of the viruses circulating each year were mapped against the Southern Hemisphere vaccine strain for the respective year, and variant analysis was conducted using the CLC Genomics Workbench. Viruses collected in 2015 were mapped to A/Switzerland/9715393/2013 (EPI_ISL_166310), those collected in 2016 and 2017 were mapped to A/Hong Kong/4801/2014 (EPI_ISL_270160), those collected in 2018 were mapped to A/Singapore/INFIMH-16-0019/2016 (EPI_ISL_330262), and those collected in 2019 were mapped to A/Switzerland/8060/2017 (EPI_ISL_332305). In addition, viruses in 2019 were further mapped to the 2020 Southern Hemisphere vaccine strain, A/South Australia/ 34/2019 (EPI_ISL_413291). Mutational analysis was performed on the six longest virus genes (PB2, PB1, PA, HA, NP, and NA) of eight segments to determine the difference in amino acid substitutions between each year’s vaccine strain and circulating strains.

### 2.10. Subclade Classification and Reassortment Analysis Using Phylogenetic Trees

Subclade classification for the rest of the seven segments other than HA, PB2, PB1, PA, NP, NA, MP, and NS was implemented by phylogenetic tree analysis using the vaccine strains as reference strains for their respective subclades. MEGA software v.7.0.26 (https://www.megasoftware.net/; accessed on 12 January 2022) was used to construct maximum-likelihood trees of eight individual segments, using HKY + G, Tamura’s three-parameter, and Kimura’s two-parameter distance model (with the gamma distribution of among-site rate variation with five categories estimated from the empirical data), with 1000 bootstrap replicates. To detect reassortment, we compared the topologies of all trees of the eight segments that matched the phylogeny of the HA segment with those of other segments to identify inconsistencies arising from intra-subgroup reassortment.

### 2.11. Evolutionary Analysis and Estimation of Nucleotide Substitutions Rates

To compare the molecular evolutionary rate of each gene within a host and to determine how the rate of evolution acts on the HA gene, which evolves more rapidly than other segments, the overall rates of evolutionary change (number of nucleotide substitutions rates per site per year) were estimated using the Bayesian statistical inference approach implemented in BEAST version 1.8.4 [32]. BEAST was run for 10 million MCMC chains with sampling every 1000 generations. We used the molecular-clock method to estimate the evolutionary rates of each gene. A relaxed molecular clock (uncorrelated lognormal) and HKY + G for the HA gene and Tamura 3 parameter for the NA gene were used after the selection of the best-fitting nucleotide substitution model in MEGA software v.7.0.26 (https://www.megasoftware.net/; accessed on 12 March 2022). Convergence was assessed based on an effective sample size value > 200 for each parameter in every run using Tracer version 1.7 (https://github.com/beast-dev/tracer/releases/tag/v1.7.2; accessed on 12 March 2022). To calibrate the molecular clock, prior evolutionary rates of 4.84 × 10^−3^ substitutions/site/year (95% highest posterior density (HPD), 5.38 × 10^−3^ to 4.32 × 10^−3^) were used for the HA segment as well as the other segments [33] except for NS. For NS, data from a previously published paper [34] were used in this analysis. This value was used as the estimation method in the previous study and was similar to that used in our analysis, especially for the whole-genome analysis of the influenza A/H3N2 virus. For the whole-genome dataset in Myanmar, the mean rate of nucleotide substitution was estimated using a coalescent reassortant constant population model implemented in BEAST software v.1.8.4 [32] with the following parameters: GRT + G + I substitution model, strict clock, and exponential prior reassortment rate with a mean of 0.00254 [35].

## 3. Results

### 3.1. Characterization of Influenza Virus Isolates

A total of 2058 nasal swabs were collected from outpatients with ILI from Myanmar between 2015 and 2019. Of the 1303 samples that tested positive in the rapid diagnostic test, 803 (61.6%) tested positive for influenza using qPCR. Of these 803 samples, 444 (55.3%) were classified as influenza A, with 129 (29.1%) classified as A/H3N2 and 315 (70.9%) classified as A/H1N1. The remaining 359 (44.7%) samples were classified as influenza B. These findings indicate that influenza A viruses were slightly more predominant in Myanmar during the study period.

Seventy-nine influenza A/H3N2 viral isolates were subjected to NGS molecular genetic analysis. All 79A/H3N2 consensus genomes were successfully sequenced, and those with >95% coverage were analyzed. Segment coverage generally declined as the segment size increased. However, an average mean depth of 1000 to 3000 was achieved for all segments.

### 3.2. Subclade Analysis by Amino Acid Substitutions and Phylogenetic Tree Analysis of HA

All 79 Myanmar isolates contained amino acid substitutions at L3I, N144S, N145S, F159Y, K160T, N225D, and Q311H in HA1 compared with the 2014 A/Texas/50/2012 vaccine strain. The findings indicated that 79/79 A/H3N2 viruses in Myanmar collected between 2015 and 2019 belonged to clade 3C.2a. Of the 79 isolates, analysis of amino acid substitutions in HA showed that all strains in 2015, the majority of strains in 2016, and one strain in 2018 belonged to subclade 3C.2a (*n* = 39.49%). Two strains in 2016 were in subclade 3C.2a1 (*n* = 2.3%), and all strains except one from 2017 to 2019 were in subclade 3C.2a1b (*n* = 38.48%). Notably, one 2018 isolate (18M141) belonged to subclade 3C.2a. Precise amino acid mutations in HA1, which define subclades compared to the 2014 A/Texas/50/2012, a Southern Hemisphere vaccine strain, are summarized in Appendix A.

Phylogenetic analysis of HA verified that all Myanmar sequences, including vaccine strains, belonged to clade 3c.2a (Figure 1A) and its subclades according to the amino acid defined in HA by the WHO (https://www.crick.ac.uk/research/platforms-and-facilities/worldwide-influenza-centre/annual-and-interim-reports; accessed on 28 April 2022).

To understand the genetic relationship between the Myanmar strains and the globally circulating strains, we performed a BLAST search for the HA gene of 79 A/H3N2 sequences and constructed a phylogenetic tree. The HA gene in Myanmar was most closely related to the strains isolated in China, Thailand, India, the USA, Australia, and Canada from 2015 to 2019 (Appendix A). To better understand the clade classification and group clustering of the viruses, a phylogenetic tree was constructed using 12 strains from India and Thailand after comparing them with the Southern Hemisphere vaccine strains of known clades recommended by the WHO. We noted that the 2015 and 2016 A/H3N2 viruses in Myanmar belonging to clades 3C.2a and 3C.2a1, respectively, were closely related to the 3C.2a clades of the 2017 Indian viruses. Subsequently, the viruses isolated from 2017 to 2019 in Myanmar clustered together with the 2017 viruses from India and the 2018 and 2019 viruses from Thailand, which belonged to clade 3C.2a1b (Appendix A).

### 3.3. Analysis of Comparison of Amino Acid Substitutions between Circulating Strains and A/Texas/50/2012 Vaccine Strain in HA

Compared with the 2014–2015 vaccine strain A/Texas/50/2012, mutations L3I, N144S, N145S, F159Y, K160T, and Q311H were detected in most samples (2015–2019) (Appendix A). Among these mutations, the prominent mutations N121K and N171K in HA1 in 2016 and E62G, N121K, T135K, K160T, and N171K between 2017 and 2019 (Appendix A) resulted in the evolution of viruses and the generation of the new clades 3C.2a1 and 3C.2a1b, respectively.

### 3.4. Comparison of Amino Acid Substitutions between Circulating Strains and Vaccine Strains in HA and Five Other Segments

We compared amino acid substitutions in HA between circulating A/H3N2 viruses isolated from 2015 to 2019 and Southern Hemisphere vaccine strains from the relevant years. A total of 13 common amino acid substitutions (L19I, S130T, A144T, S154A, G158R, N160S, S175Y, K176T, N241D, Q327H, R342K, K466R, and D505N) were observed in the HA of all the 2015 isolates in Myanmar when compared with HA gene sequences of the 2015 vaccine strain A/Switzerland/9715293/2013. Seven sporadic mutations (G21R, D69N, S140R, P185S, K280R, G395R, and G495R) were detected in HA in some of the 2015 isolates (Table 1).

HA sequences were compared with the 2016 and 2017 vaccine strain A/Hong Kong/4801/2014 for the 2016 and 2017 isolates in Myanmar. Six common amino acid substitutions (S112N, K176T, N187K, P210L, I422V, and G500E) and six additional sporadic mutations (R49Q, N137K, S160G, R158G, N174H, and I258M) were observed in some of the2016 isolates. A total of 12 common amino acid substitutions (E78G, K108R, S112N, N137K, T151K, R158G, K176T, N187K, P210L, H327Q, I422V, and G500E) and an additional four sporadic mutations (G94D, D287N, P305S, and Q372L) were detected in the HA of some of the 2017 isolates in Myanmar (Table 1).

Similarly, the 2018 Myanmar isolates were compared to the 2018 vaccine strain of Southern Hemisphere A/Singapore/ INFIMH-16-0019/2016. Eight common amino acid substitutions (I64R, E78G, K108R, T144A, T151K, V325I, H327Q, and E495G) were detected in all Myanmar isolates in 2018. Twelve additional sporadic mutations (T176R, S140R, N174H, S160K, K187N, R277Q, V422I, A546V, Q213R, K223R, S235F, and V363M) were found in some of the 2018 isolates. Compared with the 2019 vaccine strain A/Switzerland/8060/2017, a total of 19 common amino acid substitutions (A16T, R49Q, E78G, K108R, S112N, N137K, T144A, K147T, T151K, S153F, A154S, K158G, N187K, F209S, Q277R, H327Q, I422V, G500E, and Q517R) were observed. Seven additional sporadic mutations (K99E, Y110N, K205R, V363M, I538M, V545I, and A546V) were detected in the HA of some of the Myanmar isolates in 2019 (Table 1). In general, the number of amino acid mutations observed between the circulating strains and the following year’s vaccine strains was lower, indicating that the circulating strains were similar to the vaccine strains in the following year.

We limited our mutation analysis to the six longest viral genes (PB2, PB1, PA, HA, NP, and NA; Appendix A) because the two shortest influenza genes (MP and NS) have alternatively spliced and partially overlapping reading frames that complicate the annotation of mutations. Among the detected mutations, the isolates of the 3C.2a1b clade also contained specific amino acid mutations at P126L, K220N, and V303I in the NA (Appendix A), S107N in PB2 (Appendix A), and K158R in the PA (Appendix A) of H3N2 viruses in Myanmar between 2017 and 2019.

### 3.5. Analysis of Whole-Genome Sequence Data and Identification of Reassortments

The maximum-likelihood phylogenetic reconstruction of HA revealed the presence of three distinct subclades (3C.2a, 3C2a1, and 3C.2a1b) within the A/H3N2 subtype in Myanmar (Figure 1A). To determine the evolutionary relationship between gene segments, phylogenetic trees for the other seven segments were generated, and the positions of the subclades were compared among the eight segments (Figure 1A–H). The HA segment (Figure 1A) of subclade 3C.2a1 viruses that appeared in 2016 (pink in Figure 1) was closer to the newer subclade 3C.2a1b viruses (orange letters) that appeared from 2017 to 2019. The HA segment revealed a topology that was generally consistent with that of the phylogenetic tree generated using the PA, MP, and (despite the low resolution) NS segments (Figure 1D,G,H). In contrast, in four of the eight gene segments (PB2, PB1, NP, and NA) (Figure 1B,C,E,F), subclade 3C.2a1 viruses (colored pink) were closer to the older 3C.2a clade viruses (green letters) from 2015 to 2016. These findings suggest that the 3C.2a1 subclade viruses were minor reassortants between 3C.2a and 3C.2a1b.

Notably, one strain (18M 141; blue circled) collected in 2018 belonged to subclade 3C.2a, which circulated mainly from 2015 to 2016. This differs from other 2018 strain clusters. This finding suggests that this strain was a holdover from the previous 3C.2a clade viruses, which appeared a few years later.

### 3.6. Nucleotide Substitution Rates Using the Bayesian MCMC Method

Nucleotide substitutions in the whole-genome of Myanmar H3N2 viruses evolved at mean rates of 1.95 × 10^−3^ mutations/site/year (95% highest posterior density, 1.68 × 10^−3^ to 2.24 × 10^−3^) across eight genes. The mean rates of nucleotide substitution of the individual segments varied from 1.64 × 10^−3^ to 3.37 × 10^−3^ nucleotide substitutions/site/year, with the highest rates of nucleotide substitution for the major glycoproteins HA (3.37 × 10^−3^ substitutions/site/year) and NA (2.89 × 10^−3^ substitutions/site/year) and the lowest rate for the NS segment. Multiple evolutionary rates carried by other segments showed different rates of nucleotide substitution. The PB2 and the PB1 segments were evolving at similar rates of 1.95 × 10^−3^ substitutions/site/year and 1.64 × 10^−3^ substitutions/site/year. However, the corresponding rates of change of the other proteins varied considerably; the PA segment (2.20 × 10^−3^ substitutions/site/year) displayed a higher rate than that of PB1, NP (1.84 × 10^−3^ substitutions/site/year), MP (1.80 × 10^−3^ substitutions/site/year), and NS1 (4.19 × 10^−5^ substitutions/site/year) (Table 2).

## 4. Discussion

To our knowledge, this is the first comprehensive whole-genome sequencing study of influenza A/H3N2 viruses in Myanmar. We compared genetic differences between circulating strains and each season’s Southern Hemisphere vaccine strains using the whole genome sequences. The ladder-like phylogenies in Myanmar represent the antigenic drift of H3N2, as the viruses differed from the vaccine strain in each season. This finding indicated that the vaccine strains did not match those of the circulating strains in Myanmar. Subclade 3C.2a1 viruses evolved and displayed minor reassortants in 2016. The viruses completely changed into the newer 3c.2a1b subclades from 2017 to 2019. Additionally, we calculated the average genetic evolution rate between the eight segments of the A/H3N2 viruses. The number of mutations and the rate of nucleotide substitution were the highest in the HA gene, whereas parallel evolution varied considerably between other segments, with distinct relationships forming in different H3N2 virus subclades arising from unique evolutionary paths.

All eight segments of H3N2 viruses in Myanmar from 2015 to 2019 were different from the vaccine strains of the respective years but were similar to those detected in the following season. Thus, the vaccine strains were selected one year before the circulating virus [36]. Weekly surveillance data from 10 Asia countries from 2006 to 2011 indicated that most tropical and subtropical countries in Southeast Asia were associated with discrete seasonality, with peak periods of influenza activity between June/July and October; thus, they exhibited earlier influenza activity peaks than temperate climate countries [37]. The current vaccination schedules used for the Northern Hemisphere vaccines were suboptimal, and conducting vaccination from April to June each year that complies with the Southern Hemisphere vaccines (i.e., prior to the peak circulation of influenza viruses) may be beneficial for most tropical countries in Southeast Asia [37].

Among the eight segments, the highest number of mutations was found in HA, with a range of 20, 20, and 27 detected in the 2015, 2018, and 2019 isolates, respectively. The lowest number of mutations was found in HA, with a range of 12 to 16 that were detected in the 2016 and 2017 isolates in Myanmar, respectively. A similar finding revealed that circulating strains diverged from the WHO-recommended season vaccine strains, which were reported in the HA sequences in Kenya during the 2007–2013 seasons [38], all eight sequences in Thailand during the 2018–2019 seasons [39], and HA sequences in India during the 2017 season [40].

To determine the evolutionary relationship between the gene segments, eight individual phylogenetic trees of the whole genome were generated. The full genome phylogeny in Myanmar suggested the presence of minor reassortant viruses in the 3C.2a1 subclade during the 2016 season. In the HA, PA, and MP segments, these 3C.2a1 subclade viruses were closer to the newer subclade 3C.2a1b viruses that appeared from 2017 to 2019. In the PB2, PB1, NP, and NA segments, these viruses were closer to the older 3C.2a clade viruses that appeared from 2015 to 2016. Similar reassortant viruses, but in a different subclade (3C.2a2), were reported in the USA in the 2017–2018 season, in which reassortment events occurred in both HA and PB1 segments [41]. One strain (18M141) in the 2018 season was very similar to the old subclade (3C.2a) in the 2015 season, presumably indicating that this strain orignated from the old ancestor strain, having survived from 2015 to 2018.

The HA protein plays a key role in determining the virulence of influenza viruses and is closely related to the prevalence of influenza. This protein is the molecular basis for the antigenic variation of influenza viruses [42]. To compare the HA genome sequences of 3C.2a virus detected in Myanmar from 2015 to 2019 with those observed in foreign countries, we performed a BLAST search. The HA genome sequences of the Myanmar isolate clustered with isolates from other areas, including strains from China, Thailand, India, the USA, Australia, and Canada during the same period [39,40]. In our previous study, influenza A/H3N2 viruses were the predominant viruses circulating in Myanmar in 2010 and 2015. Phylogeographic analysis revealed that these viruses originated in Europe and disseminated to various countries via Australia [9], presumably indicating that even now, the influenza virus in Myanmar is closely related to those in neighboring Asian countries and Australia and remotely to those in Europe or the Americas.

Compared with the 2014–2015 vaccine strain A/Texas/50/2012, L3I, N144S, N145S, F159Y, K160T, and Q311H mutations were detected in most of the samples (2015–2019) (Appendix A). Among these mutations, the prominent mutations N121K and N171K in HA1 in 2016 and E62G, N121K, T135K, K160T, and N171K between 2017 and 2019 resulted in the evolution of the viruses and the generation of the new clades 3C.2a1 and 3C.2a1b, respectively. Similar findings have been reported globally, where phylogenetic analysis of HA genes from viruses showed diversity and sequences belonging to clades 3C.2a and 3C.3a. Moreover, 3C.2a viruses with similar mutations predominated in the 2016 isolates in Greece [43], 2017 isolates in India [40], and 2018–2019 isolates in Thailand [39]. Among the detected mutations, T135K and N171K substitutions were located within antigenic sites. Thus, they potentially affected viral antigenicity as they resulted in the loss of *N*-linked glycosylation sites, which can affect the antigenic and other properties of viruses [43]. Furthermore, Yu et al. and Zost et al. also found that the generation of protective antibody responses against an H3N2 vaccine strain occurred during the 2016–2017 H3N2 influenza season owing to the substitutions of K160T, resulting in a new glycosylation site in the HA protein, which affected antigenicity [27,44]. The K160T mutation was found in all Myanmar isolates, indicating that this mutation is related to antigenicity, which may affect the binding ability of antibodies. These substitutions most likely resulted in antigenic drift and the emergence of variant viruses. Furthermore, these data suggest that HA has powerful mutation potential, possibly as a consequence of this history of strong and repeated selection for antigenic escape.

For the NA sequences of H3N2 viruses in Myanmar between 2017 and 2019, V303I (observed in 2017, 2018, and 2019 Myanmar isolates clade 3C.2a1b) has been reported in A/H3N2 viruses with a low resistance to neuraminidase inhibitors [45]. Unfortunately, a susceptibility assay for neuraminidase inhibitors was not performed in this study. Moreover, we did not detect mutations conferring resistance to neuraminidase inhibitors, such as E119V, D151E, I222V, R224K, E276D, N249S, R292K, and R371K [46] in the NA gene segments of the Myanmar isolates. This finding indicates that the viruses circulating in Myanmar between 2015 and 2019 had difficulty acquiring resistance to neuraminidase inhibitor drugs. This may be owing to the limited access to antiviral drug prescriptions of clinicians in Myanmar. Similarly, no community-circulating strains resistant to neuraminidase inhibitors were reported in Thailand [39] and Canada [46] during the same period. Simon et al. [47] demonstrated that the two most significant mutations (V263I in the NA sequence and K196E in the NS sequence) were associated with severity and co-occurred only in viruses from the 3C.2a1 clade. We did not detect any of these mutations in the present study. Thus, the isolated samples collected between 2015 and 2019 in Myanmar may not have been associated with severe infection. We also found additional mutations in the NA segment, such as K75R, I212V, and 176I. However, the contribution of these mutations to resistance to neuraminidase inhibitors and their impact on viral fitness has not been reported.

In this study, the amino acid substitutions of K158R in the PA segment and S107N in the PB2 segment were observed in Myanmar isolates, similar to those in Thailand [39]. These substitutions may influence the virulence of the influenza virus. There are no documented reports, and further investigations on the transmission of these variants are needed.

We used a molecular clock approach to calculate the evolutionary rates of all the influenza A/H3N2 virus isolates in Myanmar. A comparison of HA with PB2, PB1, PA, NP, MP, and NS revealed that HA showed the highest rate of amino acid substitution. The evolutionary rates of PB2, PB1, PA, HA, and NA were higher than those of other segments in H3N2 viruses. One might expect PB2, PB1, PA, and NP to co-evolve because of the functions of the proteins they encode. The polymerase subunits PB2, PB1, and PA form a supramolecular complex around each NP [48]. The demonstration that the NA segment preferentially colocalizes with the viral RNA of one viral ribonucleic protein component supports the possibility that parallel evolution of NA with PB2, PB1, PA, and NP could also be driven by RNA–RNA interactions, as they are involved in gene transcription and replication [49]. Moreover, the NA and PA segments displayed higher evolutionary rates than the other segments. Our findings are similar to those of other studies [33,50,51]. Although the differences were minimal, PA had the highest rate of amino acid substitutions in the polymerase complex proteins. The differences between NS1 and the other segments were even larger, showing the lowest rate of amino acid substitutions in the NS1 segment. The full genome of Myanmar H3N2 viruses evolved at a mean rate of 1.95 × 10^−3^ mutations/site/year, indicating a slower evolution than viruses isolated in Uganda (2.76 × 10^−3^ mutations/site/year; 95% HPD, 2.54 × 10^−3^ to 2.99 × 10^−3^) across eight genes in African viruses between 2010 and 2018 [35].

The present study has some limitations. First, the samples were collected from only two regions in Myanmar. The sequenced viruses reflected half of all A/H3N2 isolates in this study, which may not represent the genetic diversity of influenza viruses for the entire country. Second, some laboratory investigations of the HA assay and antiviral sensitivity test results were waived. Furthermore, structural modeling of the H3N2 protein structure was not possible. Third, we were not able to compare the evolutionary rates with rates from previous studies because this study provides the first evolutionary data from the whole genomes of influenza circulating in Myanmar. Fourth, there are limited data in this study regarding risk factors associated with disease severity because we only included outpatients who visited the hospitals in our study and did not conduct a population-based study. Further studies are needed to investigate the possible mechanisms underlying the genetic variants in the Myanmar isolates and viral replication or the role of the host response in infections with mutant viruses.

Influenza A/H3N2 viruses undergo continuous evolution, and different lineages have appeared [34]. The genome-wide relationship was examined by investigating the influenza A/H3N2 virus, which continuously changes across varying times and locations owing to drift in its phylogeny. To the best of our knowledge, this is the first molecular study of whole-genome influenza sequences in Myanmar that focuses on the evolutionary pattern of the genes and the relationship between genetic evolution and different segments. Whole-genome sequencing enables the use of powerful phylogenetic methods that substantially improve phylogenetic classification. This study provides more information for national influenza prevention and control programs regarding the timing, circulation patterns, and transmission of seasonal epidemics.

## Figures and Tables

**Figure 1 viruses-14-02414-f001:**
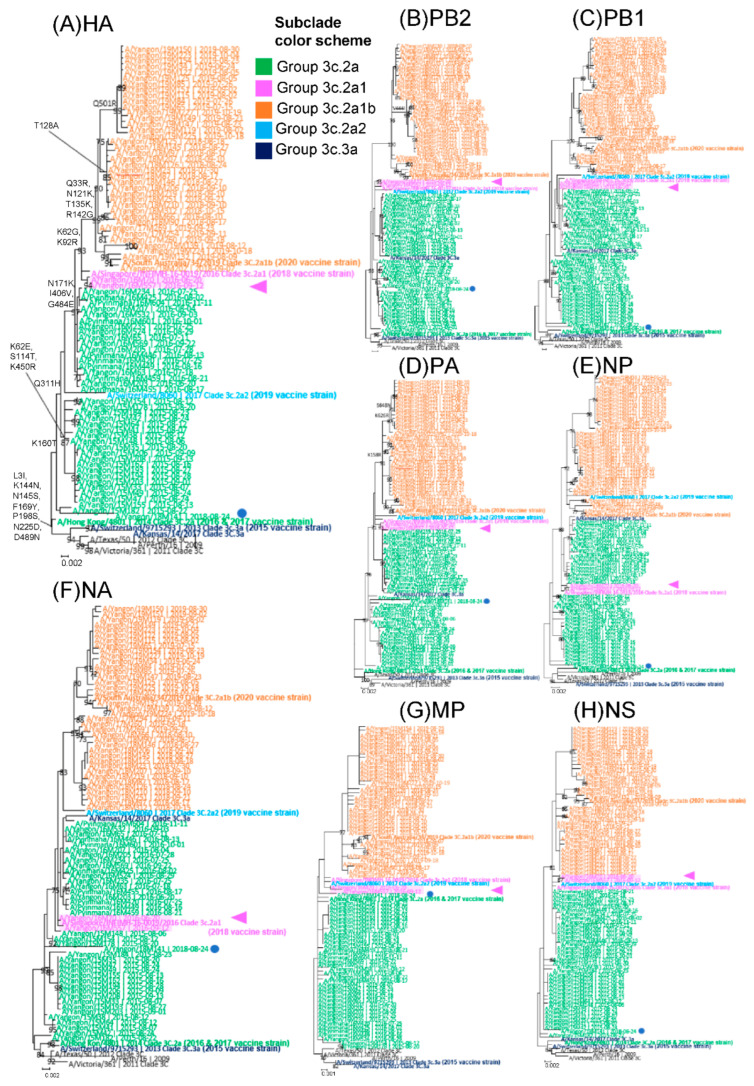
Maximum-likelihood phylogenetic trees of all eight segments of influenza A/H3N2 viruses circulating in Myanmar and comparison of sequences from 79 strains isolated in Myanmar between 2015 and 2019 and to the Southern Hemisphere vaccine strains of known clades recommended by the WHO. Southern Hemisphere vaccine strains from 2015 to 2019 are indicated by bold letters according to clade classification by the WHO. (**A**) HA, (**B**) PB2, (**C**) PB1, (**D**) PA, (**E**) NP, (**F**) NA, (**G**) MP, and (**H**) NS segments. Amino acid substitutions compared to A/Texas/50/2012 were indicated in HA. The two strains that belonged to 3c.2a1 are highlighted in pink and denoted by pink triangles to clarify the topologies concerning 3c.2a and 3c.2a1b viruses. One revival strain, A/Yangon/18M141/2018, is marked with blue circles. The phylogenetic tree was inferred by the maximum-likelihood method using 1000 bootstrap replicates implemented in MEGA v.7.0.26. Branch values > 70% are indicated at the nodes.

**Table 1 viruses-14-02414-t001:** List of mutation differences in HA gene between Southern Hemisphere vaccine strains and Myanmar 2015–2019 viruses.

Representative Strain	Clade		Amino Acid Mutations in Each Nucleotide Position																		
**2015**		**19**	21	69	**130**	140	**144**	**154**	**158**	**160**	**175**	**176**	185	**241**	280	**327**	**342**	395	**466**	495	**505**
A/Switzerland/9715293/2013 ^a^	3C.3a	**L**	G	D	**S**	S	**A**	**S**	**G**	**N**	**S**	**K**	P	**N**	K	**Q**	**R**	G	**K**	G	**D**
2015 Myanmar viruses	3C.2a	**I**	R	N	**T**	R	**T**	**A**	**R**	**S**	**Y**	**T**	S	**D**	R	**H**	**K**	R	**R**	R	**N**
**2016**		49	**112**	137	158	160	174	**176**	**187**	**210**	258	**422**	**500**								
A/Hong Kong/4801/2014 ^b^	3C.2a	R	**S**	N	R	S	N	**K**	**N**	**P**	I	**I**	**G**								
2016 Myanmar viruses	3C.2a and 3C.2a1	Q	**N**	K	G	G	H	**T**	**K**	**L**	M	**V**	**E**								
**2017**		**78**	94	**108**	**112**	**137**	**151**	**158**	**176**	**187**	**210**	287	305	**327**	372	**422**	**500**				
A/Hong Kong/4801/2014 ^b^	3C.2a	**E**	G	**K**	**S**	**N**	**T**	**R**	**K**	**N**	**P**	D	P	**H**	Q	**I**	**G**				
2017 Myanmar viruses	3C.2a1b	**G**	D	**R**	**N**	**K**	**K**	**G**	**T**	**K**	**L**	N	S	**Q**	L	**V**	**E**				
**2018**		**64**	**78**	**108**	140	**144**	**151**	160	174	176	187	213	223	235	277	**325**	**327**	363	422	**495**	546
A/Singapore/INFIMH-16-0019/2016 ^c^	3C.2a1	**I**	**E**	**K**	S	**T**	**T**	S	N	T	K	Q	K	S	R	**V**	**H**	V	V	**E**	A
2018 Myanmar viruses	3C.2a and 3C.2a1b	**R**	**G**	**R**	R	**A**	**K**	K	H	R	N	R	R	F	Q	**I**	**Q**	M	I	**G**	V
**2019**		**16**	**49**	**78**	99	**108**	110	**112**	**137**	**144**	**147**	**151**	**153**	**154**	**158**	**187**	205	**209**	**277**	**327**	363
A/Switzerland/8060/2017 ^d^	3C.2a2	**A**	**R**	**E**	K	**K**	Y	**S**	**N**	**T**	**K**	**T**	**S**	**A**	**K**	**N**	K	**F**	**Q**	**H**	V
2019 Myanmar viruses	3C.2a1b	**T**	**Q**	**G**	E	**R**	N	**N**	**K**	**A**	**T**	**K**	**F**	**S**	**G**	**K**	R	**S**	**R**	**Q**	M
		**422**	**500**	**517**	538	545	546														
		**I**	**G**	**Q**	I	V	A														
		**V**	**E**	**R**	M	I	V														

^a^ Vaccine strain for 2015, ^b^ vaccine strain for 2016 and 2017, ^c^ vaccine strain for 2018, and ^d^ vaccine strain for 2019. Amino acid substitutions in bold represent common mutations for all isolates in each season, whereas those in normal font represent additional sporadic mutations observed in some of the relevant season strains.

**Table 2 viruses-14-02414-t002:** Mean rates of nucleotide substitutions for all segments of A/H3N2 viruses circulating in Myanmar between 2015 and 2019.

Segment	Mean Rate of Substitution (Substitution/Site/Year)	
	Mean	95% Highest Posterior Density
PB2	1.95 × 10^−3^	1.42 × 10^−3^ to 2.50 × 10^−3^
PB1	1.64 × 10^−3^	1.11 × 10^−3^ to 2.18 × 10^−3^
PA	2.20 × 10^−3^	1.57 × 10^−3^ to 2.79 × 10^−3^
HA	3.37 × 10^−3^	2.52 × 10^−3^ to 4.21 × 10^−3^
NP	1.84 × 10^−3^	1.31 × 10^−3^ to 2.44 × 10^−3^
NA	2.89 × 10^−3^	2.13 × 10^−3^ to 3.71 × 10^−3^
MP	1.80 × 10^−3^	1.06 × 10^−3^ to 2.67 × 10^−3^
NS	4.19 × 10^−5^	3.25 × 10^−5^ to 5.16 × 10^−5^

## Data Availability

The whole-genome nucleotide sequences generated in this study have been deposited in the Global Initiative on Sharing All Influenza Data (GISAID) EpiFlu database with accession numbers EPI_ISL_12870031–EPI_ISL_12995850.

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
