# Peer review of "Evolutionary Dynamics of Whole-Genome Influenza A/H3N2 Viruses Isolated in Myanmar from 2015 to 2019"

_viruses, 2022, doi:10.3390/v14112414_

Round 1

Reviewer 1 Report

This is a really well-designed well-conducted study with strong-evidence and pivotal results from public health point of view and deep insight for surveillance and prepardness porpouse.

Nothing to add.

Reviewer 2 Report

This article is about the Evolutionary Dynamics of Whole-Genome Influenza A/H3N2 Viruses Isolated in Myanmar from 2015 to 2019.In the study, the authors collected samples from hospital in Myanmer for further sequencing. The study presented a lot of genome analysis about H3N2 influenza viruses. But there are some questions need to be answered below.

In Line 55, it needs to introduce the full name of PB1.

In 2.2 viral isolation part, please write details of MDCK or MDCK-SIAT1 cells, including the cultural conditions.

In part 2.8 Evaluation of Genetic Match with Vaccine Strains , Why you just analyzed the six longest virus genes (PB2, PB1, PA, HA, NP, and NA)? The M and NS genes were not analyzed in the article.

The isolation of H3N2 influenza viruses is a challenge in viral in recent years. How did you make sure the success of viral isolation? The Hemagglutinin assay?

NGS technology can be used to the samples which are not isolated. Why you not sequence all the  129 samples of H3N2 by NGS?

The discussion of this article needs to be modified again.

The references in this study are small,please added more references.
